# Static and Resonant Properties and Magnetic Phase Diagram of LiMn_2_TeO_6_

**DOI:** 10.3390/ma15238694

**Published:** 2022-12-06

**Authors:** Tatyana Vasilchikova, Evgeniya Vavilova, Timur Salikhov, Vladimir Nalbandyan, Shanu Dengre, Rajib Sarkar, Hans-Henning Klauss, Alexander Vasiliev

**Affiliations:** 1Low Temperature Physics and Superconductivity Department, Lomonosov Moscow State University, 119991 Moscow, Russia; 2Functional Quantum Materials Laboratory, National University of Science and Technology “MISiS”, 119049 Moscow, Russia; 3Zavoisky Physical-Technical Institute, FRC Kazan Scientific Center of RAS, 420029 Kazan, Russia; 4Faculty of Chemistry, Southern Federal University, 344090 Rostov-on-Don, Russia; 5Institute for Solid State and Material Physics, Technische Universität Dresden, 01069 Dresden, Germany

**Keywords:** manganese oxide, spin order, magnetic phase diagram

## Abstract

Physical properties of the mixed-valent tellurate of lithium and manganese, LiMn_2_TeO_6_, were investigated in measurements of *ac* and *dc* magnetic susceptibility *χ*, magnetization *M*, specific heat *C_p_*, electron spin resonance (ESR), and nuclear magnetic resonance (NMR) in the temperature range 2–300 K under magnetic field up to 9 T. The title compound orders magnetically in two steps at *T*_1_ = 20 K and *T*_2_ = 13 K. The intermediate phase at *T*_2_ < *T* < *T*_1_ is fully suppressed by magnetic field *µ*_0_*H* of about 4 T. Besides magnetic phases transitions firmly established in static measurements, relaxation-type phenomena were observed well above magnetic ordering temperature in resonant measurements.

## 1. Introduction

Mixed valence manganese oxides exhibit colossal magnetoresistance effect [1,2,3,4] and attract attention as prospective materials in lithium-ion industry [5,6,7,8]. Moreover, it has been found that they are capable of strong light adsorption in the solar spectrum range and may serve as magnetic sensors, spintronic, and magnetocaloric devices [9,10,11,12,13]. Along with numerous applications, manganites are quite attractive in fundamental research due to presence of spin, charge, and orbital degrees of freedom often entangled [14,15,16,17,18]. The physics of these systems relates to metal-insulator transition, competing exchange interactions and magnetic frustration [19,20,21,22].

In contrast to the widely studied Mn^3+^/Mn^4+^ mixed-valent oxides, the properties of Mn^2+^/Mn^3+^ mixed-valent materials are less explored. Although multiple compounds (starting from hausmannite Mn_3_O_4_) are known to contain simultaneously Mn^2+^ and Mn^3+^, these cations sit usually on different crystallographic sites and only rarely mix on the same site. Lithium manganese tellurate Li_2_Mn_4_Te_2_O_12_ (LiMn_2_TeO_6_ for short) belongs to this minority [23]. With average oxidation state of 2.5, Mn ions occupy four non-equivalent octahedral positions, but only two of them are definite Mn^2+^ and Mn^3+^, whereas two others are supposedly mixed Mn^2+^/Mn^3+^ sites.

Triclinic crystal structure of LiMn_2_TeO_6_ can be considered as a heavily distorted variant of the orthorhombic *Pnn2* structure of Li_2_TiTeO_6_ [24] with Li/Mn ordering on two independent Li sites. Note that Li_2_TiTeO_6_ itself is a superstructure derived from LiSbO_3_ [25]. The differences between these three structures allow considering LiMn_2_TeO_6_ as a new structural type. Ordering of cations may only destroy both glide planes of *Pnn2* structure leading to a monoclinic space group *P112* but not to the triclinic distortion since all cations in Li_2_TiTeO_6_ reside on the two-fold axes.The Jahn–Teller effect of Mn^3+^ ions and/or Mn–Mn interactions seem to be the reasons of the triclinic distortion [23].Here, we report a comprehensive study of thermodynamic and magnetic resonance properties of the lithium manganese tellurate LiMn_2_TeO_6_.

## 2. Experimental

The preparation of LiMn_2_TeO_6_ powder samples by conventional solid-state reaction and their phase analysis, redox analysis, and crystal structure determination have been described in detail earlier [23]. Here, we used samples from the same work, stored in a tightly closed container in a desiccator. Their identity was confirmed by X-ray diffraction. Thermodynamic properties of the title compound, that is the magnetization *M* and specific heat *C_p_*, were studied using various options of Quantum Design Physical Properties Measurements System PPMS-9T in the temperature range 2–300 K under magnetic field up to 9 T.

The crystal structure of LiMn_2_TeO_6_ [23] contains four different positions of manganese, as shown in Figure 1. One can find planar, chain, and dimer motives in the crystal lattice, but none of them dominates. Therefore, we cannot treat this system as purely two- or one-dimensional. The picture is complicated by the mixed valence of manganese ions. The magnetism relates to a large number of exchange interactions, different in both magnitude and sign: at least one of the bonds has an angle close to 90°, and therefore, is characterized by a ferromagnetic exchange.

Electron spin resonance (ESR) studies were carried out using an X-band ESR spectrometer “Adani” CMS 8400 (*f* ≈ 9.4 GHz, *B* ≤ 0.7 T) equipped with a low-temperature mount, operating in the range *T* = 6–450 K. The effective *g*-factors have been calculated with respect to an external reference for the resonance field. BDPA with *g*_et_ = 2.00359 has been used as a reference material.

The ^7^Li (*I* = 3/2) nuclear magnetic resonance (NMR) spectra were measured using “Tecmag” pulse solid-state NMR spectrometer at various frequencies in the range 10–110 MHz. The NMR spectra were obtained by point-by-point integration of the intensity of the Hahn echo versus magnetic field. The spin-lattice relaxation has been studied by saturation recovery pulse sequence and stimulated echo. In the NMR field sweep spectra, it was not possible to resolve the central line and quadrupole satellites. Thus, the spin-lattice relaxation time *T*_1_ was obtained from the fitting of nuclear spin-echo decay with exponential function.

## 3. Results and Discussion

### 3.1. Electron Spin Resonance

In the whole temperature range, a single Lorentzian absorption line has been observed which can be ascribed to overlapping signals from Mn^2+^ and Mn^3+^ ions (Figure 2a). Upon lowering temperature, the signal broadens and eventually disappears below 30 K. Such a signal fading implies proximity to an onset of the long-range antiferromagnetic order and opening of the energy gap for resonance excitations. The main ESR parameters as obtained from fitting in accordance with Lorenzian profile [26] are shown in Figure 2b. The halfwidth Δ*B* of the ESR signal monotonously broadens with lowering temperature indicating presence of short-range correlation effects in the magnetic subsystem of LiMn_2_TeO_6_. The effective g-factor remains temperature-independent in the range 100–450 K and its value *g* ~1.98 ± 0.02 is typical for the manganese ions in the high-spin states of Mn^2+^ and Mn^3+^ in octahedral environment. Below 100 K, the effective *g*-factor deviates from linearity, indicating the effect of short-range magnetic correlations. The temperature dependence of the integral ESR intensity *χ*_ESR_(*T*) is shown in Figure 3a.

### 3.2. Magnetization

The temperature dependence of the magnetic susceptibility *χ* = *M*/*B* in LiMn_2_TeO_6_ is shown in the left panel of Figure 3. At elevated temperatures, the *χ*(*T*) curve follows the Curie–Weiss law with addition of a temperature-independent term *χ* = *χ*_0_ + *C*/(*T*−*Θ*), where *C* = 7.35 emu/mol K is the Curie constant and *Θ* = −95 K is the Weiss temperature. The diamagnetic term *χ*_0_ = −5×10^−4^ emu/mol was found to be in agreement with the sum of Pascal’s constants [27] of the ions constituting the LiMn_2_TeO_6_ compound. The negative value of the Weiss temperature indicates the predominance of antiferromagnetic exchange while the value of Curie constant corresponds to a system with equal numbers of Mn^2+^ (spin *S* = 5/2) and Mn^3+^ (spin *S* = 2) ions at averaged *g-*factor *g* = 1.98 ± 0.02. Upon decreasing the temperature, the *χ*(*T*) dependence exhibits a sharp maximum at about *T*_N_ = 20 K, indicating the long-range antiferromagnetic order. With further cooling, an additional intense peak appears at T^*^ = 13 K which is sensitive to the measurement protocol, i.e., either zero-field cooling (ZFC) or field cooling (FC) regimes. This fact is illustrated by the inset in Figure 3a.

To further elucidate the magnetic behavior of LiMn_2_TeO_6_ at low temperatures, we measured the *M*(*T*) dependences for the LiMn_2_TeO_6_ sample at various magnetic fields up to 9 T, as shown in Figure 3b. Upon increasing the magnetic field, the position of the *T*^*^ anomaly noticeably shifts to higher temperature, while the maximum at *T*_N_demonstrates the opposite trend. At the field *B*~ 4 T, both anomalies merge into one phase boundary, and anomaly broadens with increasing external field.

It has been found that magnetization isotherms *M*(*B*) display neither hysteresis nor saturation in magnetic fields up to 9 T, as shown in Figure 4. Below *T** ~ 13 K, *M*(*B*) curves deviate upward which is expected for the spin-flop transition. Above *T** ~ 13 K, this trend changes to the opposite and *M*(*B*) curves deviate downward from linearity.

### 3.3. acMagnetic Susceptibility

*ac* susceptibility *χ_ac_* was measured in the frequency range 0.1–10 kHz. These data are shown in the left panel of Figure 5. The real part of susceptibility *χ′* shows anomalies at 13 K and 20 K, similar to that found in *dc* magnetization. The behavior of both anomalies differs with the frequency variation. The position of the peak at *T*_1_ = 20 K stays the same while the peak at *T*_2_ = 13 K shifts to higher temperatures with increasing frequency. In the whole temperature range, the imaginary part *χ*″ is close to zero.

It is worth noting that the behavior of both anomalies with frequency variation is significantly different (inset in Figure 5a). The position of the peak at *T*_1_ remains the same, and this peak can be related to establishment of long-range order. The peak at *T*_2_ is frequency sensitive which is inherent to cluster spin glasses (see Appendix A). The shifts of *T*_1_ and *T*_2_ under external magnetic field agree with static thermodynamic data, as shown in Figure 5b. An increase of the magnetic field above 4 T leads to the shift of the position of the merged anomaly to higher temperatures.

### 3.4. Specific Heat

The temperature dependence of specific heat *C_p_* in LiMn_2_TeO_6_ at *B* = 0 T shown in the left panel of Figure 6 is in good agreement with *dc* magnetic susceptibility measurements showing two distinct anomalies at *T*_1_ and *T*_2_.

The jump of specific heat at *T*_1_ equals ∆*C_m_* = 18.7 J/mol K, as shown in the lower inset to Figure 6a. This value is two times lower than predicted in the mean field theory ∆*C_theor_*≈ 38.8 J/mol K [28]. Magnetic entropy ∆*S_m_* is also shown in this inset. It saturates at ~50 K reaching about 20 J/mol K. This value is again markedly lower than the magnetic entropy change expected from the mean-field theory ∆*S_theor_* = 28.3 J/(mol K) [28]. Overall, these data signal the formation of the short-range correlation regime in LiMn_2_TeO_6_ well above the Néel temperature.

The temperature dependences of specific heat *C*_p_(*T*) taken at various magnetic fields in LiMn_2_TeO_6_ are shown in Figure 6b. As is the case of magnetization measurements, the application of a magnetic field slightly shifts downward the position of anomaly at *T*_1_ and shifts upward the anomaly at *T*_2_.

### 3.5. Nuclear Magnetic Resonance

To study in detail the low energy spin dynamics at the microscopic level in the region of fraction of micro-eV energy scales, we have carried out ^7^Li NMR investigations. The crystal structure suggests the presence of Li in two non-equivalent sites in LiMn_2_TeO_6_ with different Mn environments. Therefore, the observed ^7^Li NMR spectrum is a superposition of the contributions from all lithium nuclei in powder specimen. Insight into the dynamic spin correlations is provided by the ^7^Li spin-lattice relaxation rate R_1_measurements. The low-temperature part of the T-dependence of the spin-lattice relaxation measured in two different external fields is shown in the left panel of Figure 7. The peaks in the temperature dependence of the relaxation correspond to the phase transition temperatures, which is about 18 K at 6.6 T external field, and 20 K at 1.8 T. Above the magnetic phase transition, a critical increase in the relaxation rate is observed, which at 1.8 T is well described by the critical exponent p ≈ 0.48. This value is close to the predictions of the mean field theory for a three-dimensional magnet [29]. At 6.6 T, an extended region of continuous increase of the relaxation rate with cooling is observed above the peak, indicating development of a very slow dynamics and strong spin correlations. 

One can compare the local static and dynamic susceptibility obtained from NMR data with the bulk static susceptibility. The nuclear spin lattice relaxation in magnets is usually governed by magnetic fluctuations in the electronic spin system and *R*_1_*T*^−1^ is proportional to the imaginary part of the local dynamic susceptibility *χ*″ [30]:(1)$${\left(T_{1}T\right)}^{-1}\propto \sum_{q}A_{⊥}^{2}\left(\vec{q},\omega \right)\cdot \frac{\chi^{\prime \prime }\left(\vec{q},\omega \right)}{\omega }$$

Here, *A* is the *q*-dependent hyperfine constant, *q* is the wave vector and ω is the Larmor frequency. As shown in Figure 7, the local dynamic spin susceptibility *R*_1_*T*^−1^ probed by NMR at the external field of 6.6 T is proportional to the bulk static susceptibility in the temperature range 90–250 K manifesting the paramagnetic regime of the electron spin system. Below 90 K, the dependence deviates from this linearity indicating the slowing down of spin fluctuations and development of correlations. 

The establishment of the antiferromagnetic order is reflected in the NMR spectra, which acquire a step-like shape that is specific for powders in such cases. The rectangle components of such a spectrum correspond to magnetically non-equivalent positions of lithium in an ordered state. The shape of the individual rectangles can be described by the equation [31]:(2)$$f\left(H,H_{A},H_{0}\right)\propto \frac{1}{4H_{A}}\left|1+\frac{H_{0}^{2}-H_{A}^{2}}{H^{2}}\right|$$
for |*H*_0_ − *H_A_*| ≤ *H* ≤ *H*_0_ + *H_A_*. Here, *H*_0_ = *γ*_n_*ω*_L_, *H_A_* is a local internal field, *ω*_L_ is the paramagnetic Larmor frequency, and *γ*_n_ is a nuclear gyromagnetic ratio. For proper interpretation of the results, a correction of the spectrum intensity to the magnitude of the measurement fields was made. The resulting spectra obtained at 12 K are shown in Figure 8 by green color. All spectra contain the narrow gaussian contribution at *H_G_* = *H*_0_ + 0.11 T with the width ~0.3 T and the intensity of about 3–4% of total intensity of the spectra (dark grey line) which apparently refers to a small fraction of the diamagnetic impurity.

Subtracting this Gaussian component from the experimental spectrum, one can obtain the ^7^Li NMR spectrum in LiMn_2_TeO_6_ (blue line). The modeling of the spectra with the 0.6 T, 1 T, and 1.7 T according to the formula (2) gives almost equal sets of the basic fitting parameters for three different lithium positions: H_0_ is a Larmor field, *H_A_*^(1)^ ≈ 0.37 T, *H_A_*^(2)^ ≈ 0.185 T, *H_A_*^(3)^ ≈ 0.1 T and relative intensities *I*^(1)^≈ 50%, *I*^(2)^≈ 30%, *I*^(3)^≈ 20%. The obtained intensity ratio *I^(1)^/(I^(2)^+I^(3)^)* is close to the ratio of the filling of structural positions (Li1:Li2 ~ 5:4). It indicates the appearance of the magnetically nonequivalent atoms in at least one of the structural positions. A strong increase in the external field up to 6.6 T leads to a partial tilting of the manganese spins along the field direction that distorts the shape of the spectrum and, strictly speaking, makes Equation (2) inapplicable. Using it formally, we can get the following simulation parameters: *H_A_*^(1)^ ≈ 0.32 T, *H_A_*^(2)^ ≈ 0.175 T and *H_A_*^(3)^ ≈ 0.095T, *I*^(1)^≈ 50%, *I*^(2)^≈ 27%, *I*^(3)^≈ 23%, but central field of each rectangles is no more *H*_0_ = 6.648 T but it shifts steadily to higher values: *H*_0_^(1)^ ≈ 6.67 T, *H*_0_^(2)^ ≈ 6.68 T, and *H*_0_^(3)^ ≈ 6.7 T. Such a different shift of the zero field on different lithium positions indicates the presence of a component of the internal field, collinear to the external one, caused by the tilting of spins in AFM sublattices in a strong external field. The absence of any changes in the spectral shape, except for those caused by gradual tilting of spins in a magnetic field, allows to attribute all these spectra to the same type of magnetic structure and the same magnetic phase.

The temperature transformation of the NMR spectrum in the ordered state obtained at 1.8 T is shown in Figure 9. The shape of the NMR spectrum changes with temperature. The Néel temperature is clearly seen in the line width that dramatically increases below 20K. A pronounced step-like profile is present at temperatures of 12.5 and 10 K. At 15.5 K, the spectrum structure is less resolvable and the width is smaller. At low temperatures, the line is smoothed, especially on the high-field shoulder, and, at the same time, has a significantly large width.

## 4. Discussion

The static magnetic properties of LiMn_2_TeO_6_ are firmly established in measurements of *dc* magnetic susceptibility, magnetization, specific heat. Summarizing the experimental results, the magnetic phase diagram for LiMn_2_TeO_6_ has been established, as shown in Figure 10. At lowering temperature, the title compound orders antiferromagnetically at *T*_1_ = 20 K and experiences second magnetic phase transition at *T*_2_ = 13 K. The intermediate phase at *T*_2_ < *T* < *T*_1_ is fully suppressed by an external magnetic field of 4 T. Such behavior is frequently observed in compounds which experience consecutive incommensurate and commensurate magnetic orders. The antiferromagnetic type of ordering at both *T* < *T*_2_and *T*_2_ < *T* < *T*_1_ is confirmed by the rectangular shape of the ^7^Li NMR line.

The dynamics of the spin system was examined by NMR and EPR. LiMn_2_TeO_6_ is characterized by an extended region of developed correlations above *T*_N_. X-band ESR signal disappears at about 40 K. Dynamic susceptibility on MHz timescale measured by NMR relaxation rate shows noticeable deviations from the bulk susceptibility below ~50K. Moreover, while the T_1_^−1^(T) dependence obtained at 1.75 T is govern by 3D critical regime in the upper vicinity of T_N_, it is not the case at higher magnetic fields. Such behavior proves the development a slow Mn spin correlations well above the ordering temperature. Usually if the correlations in a magnetic system slow down with decreasing temperature, it is usually followed, almost immediately, by a static ordered or glassy regime. The wide temperature range of slow and not three-dimensional correlations indicates that magnetism of LiMn_2_TeO_6_is determined by a complex network of Mn^3+^ and Mn^4+^ exchange interactions which are of different magnitude and partially frustrated. The competition of these interactions apparently causes the appearance of an intermediate phase below the Néel temperature.

It could be of interest to compare physical properties of LiMn_2_TeO_6_ with those of its iso-elemental counterpart Li_2_MnTeO_6_ [22]. While the magnetic susceptibility of Li_2_MnTeO_6_ shows no obvious anomaly indicative of a long-range magnetic order at low magnetic fields, at high magnetic field it evidences the antiferromagnetic-type peak at about 9 K confirmed also by specific heat measurements. Furthermore, this conclusion is supported also by ^7^Li NMR data and dielectric permittivity measurements. Density functional theory calculations lead to a 120° noncollinear spin arrangement which agrees with the magnetic structure defined in neutron-diffraction measurements.

Despite similar chemical compositions of LiMn_2_TeO_6_ and Li_2_MnTeO_6_, their physical properties differ drastically due the difference of oxidation states, i.e., Mn^2+^/Mn^3+^ in the former compound and Mn^4+^ in the latter compound. Mixed valence of manganese results in the appearance of ferromagnetic double exchange on the background of antiferromagnetic superexchange in LiMn_2_TeO_6_ to be compared with purely antiferromagnetic exchange in Li_2_MnTeO_6_.

## 5. Summary

Summarizing, the competition of various exchange interactions and the presence of both divalent and trivalent manganese ions in LiMn_2_TeO_6_ leads to the two-step formation of antiferromagnetic state at *T*_1_ = 20 K and *T*_2_ = 13 K. The phase at *T*_2_ < *T* < *T*_1_ is readily suppressed by an external magnetic field *µ*_0_*H* of 4 T. At high fields, well above the Néel temperature, an extended correlation region is found, characterized by large correlation time. It can be explained by the multicomponent nature of the magnetic system, which has a complicated geometry and is partially frustrated.

## Figures and Tables

**Figure 1 materials-15-08694-f001:**
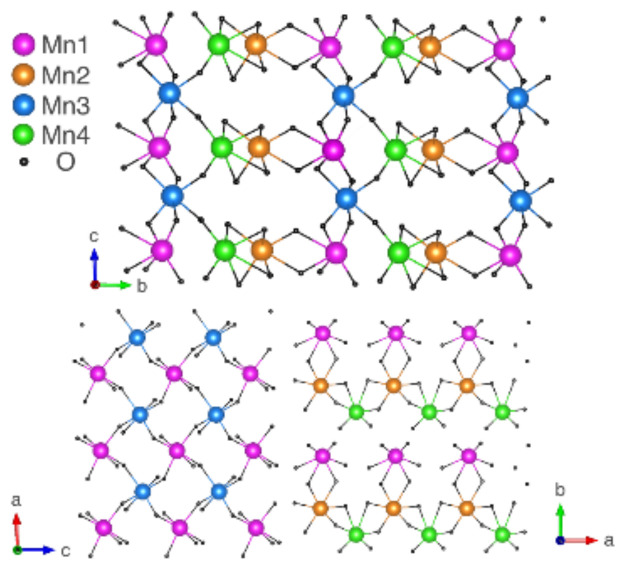
Arrangement of manganese ions in the crystal structure of LiMn_2_TeO_6_ in different planes.

**Figure 2 materials-15-08694-f002:**
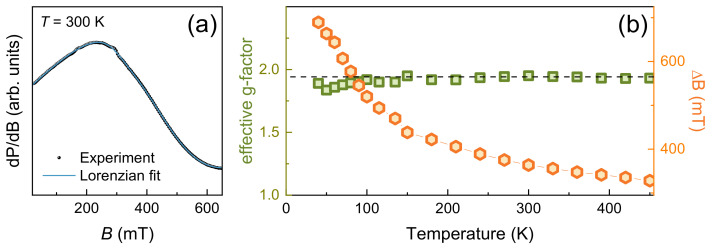
(**a**) ESR spectrum taken at room temperature. (**b**) The temperature dependences of the effective g-factor and ESR linewidth ∆*B* in LiMn_2_TeO_6_.

**Figure 3 materials-15-08694-f003:**
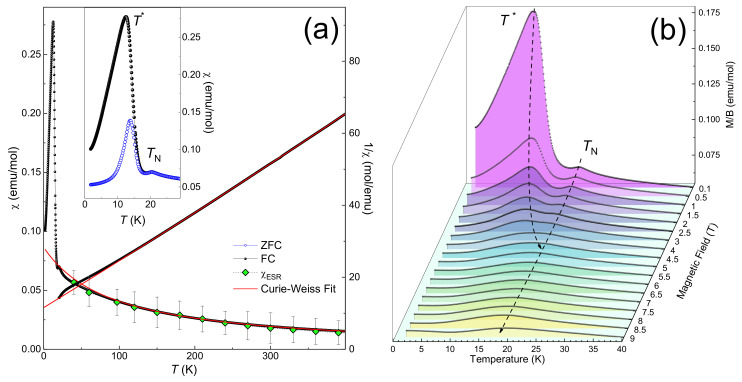
(**a**) The temperature dependences of the magnetic susceptibility recorded in ZFC (blue open symbols) and FC (black sphere) regimes at *B* = 0.1 T and the integrated ESR intensity (light green diamond) along with inverse magnetic susceptibility 1/*χ*. The red solid line represents an approximation in accordance with the Curie-Weiss law. (**b**) The *M*(*T*) curves for the LiMn_2_TeO_6_ taken at various external magnetic fields.

**Figure 4 materials-15-08694-f004:**
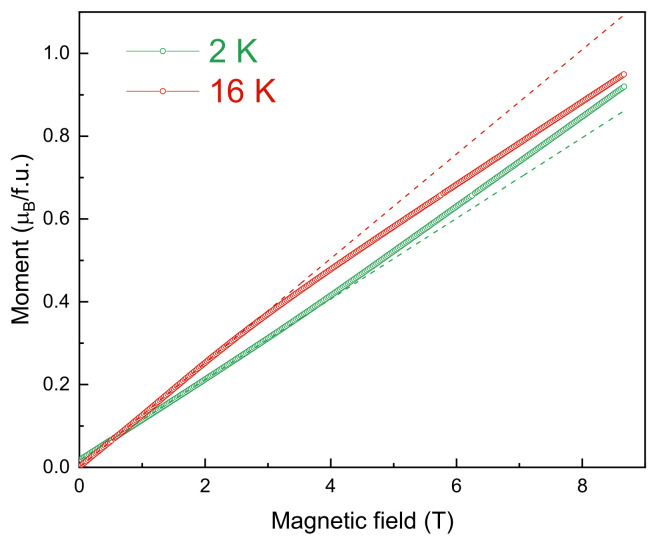
The magnetization isotherms at *T*<*T*_2_ (lower curve) and at *T*2 <*T*<*T*_1_ in LiMn_2_TeO_6_.

**Figure 5 materials-15-08694-f005:**
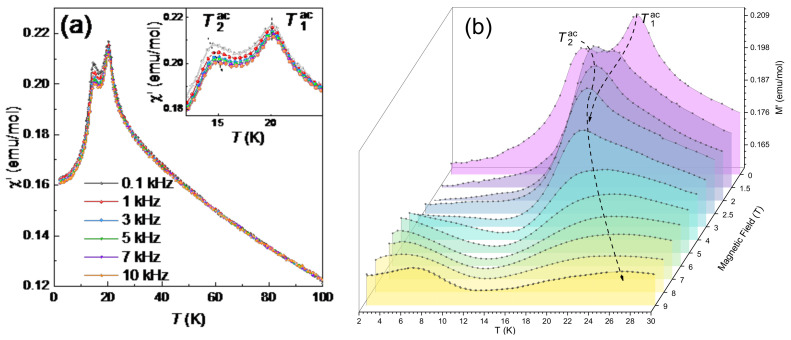
(**a**) Real χ′ part of the ac magnetic susceptibility at various frequencies. Inset: the zoomed-in region of real χ′ near transitions. (**b**) The temperature dependencies of real part χ′ of the *ac* magnetic susceptibility at various magnetic fields at *f* = 10 kHz.

**Figure 6 materials-15-08694-f006:**
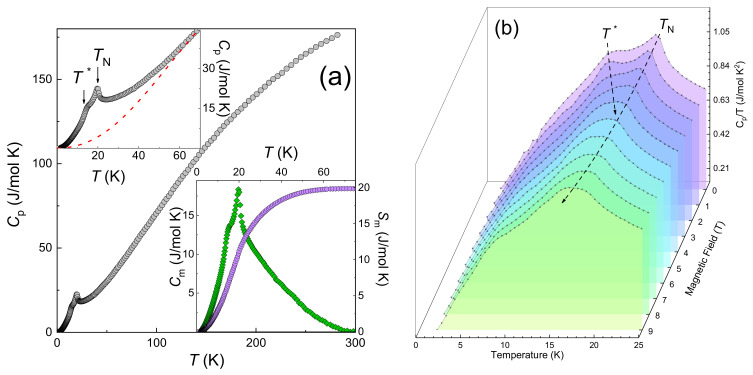
(**a**) Temperature dependence of the specific heat in LiMn_2_TeO_6_ in zero magnetic field. Upper inset highlights the two-step transition at low temperatures. Dash line represents the lattice specific heat within frames of Debye model. Lower inset shows magnetic specific heat *C*_m_(T) and magnetic entropy ∆*S*_m_. (**b**) Temperature dependences of specific heat *C*_p_(*T*) in LiMn_2_TeO_6_ at various magnetic fields.

**Figure 7 materials-15-08694-f007:**
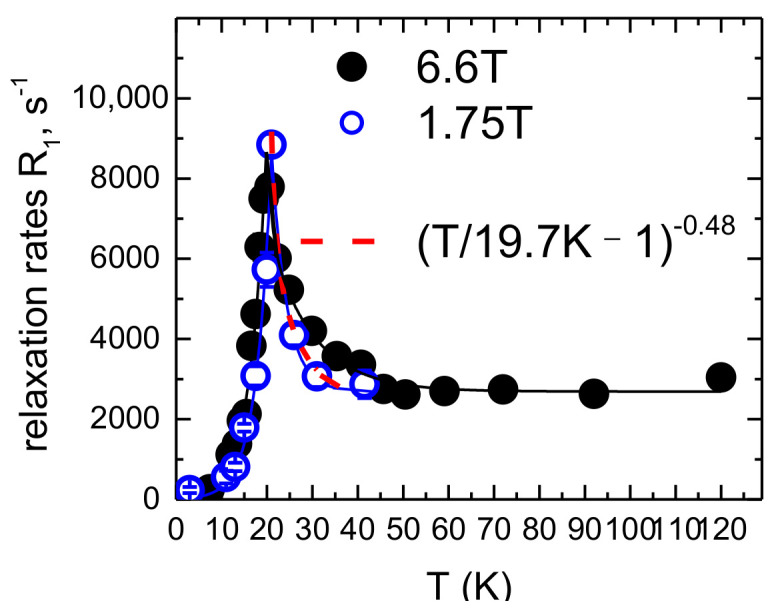
Temperature dependence of the spin-lattice relaxation rate R_1_ in LiMn_2_TeO_6_ at two external fields. Solid circles correspond to 6.6 T, open circles to 1.75 T. Dashed line is a critical exponent for 1.75 T (see text).

**Figure 8 materials-15-08694-f008:**
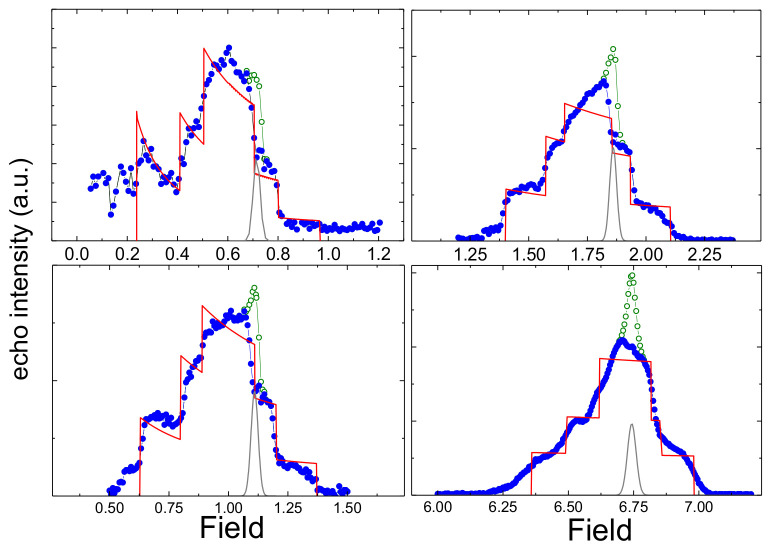
NMR spectra at 12 K obtained in LiMn_2_TeO_6_ under different external field.

**Figure 9 materials-15-08694-f009:**
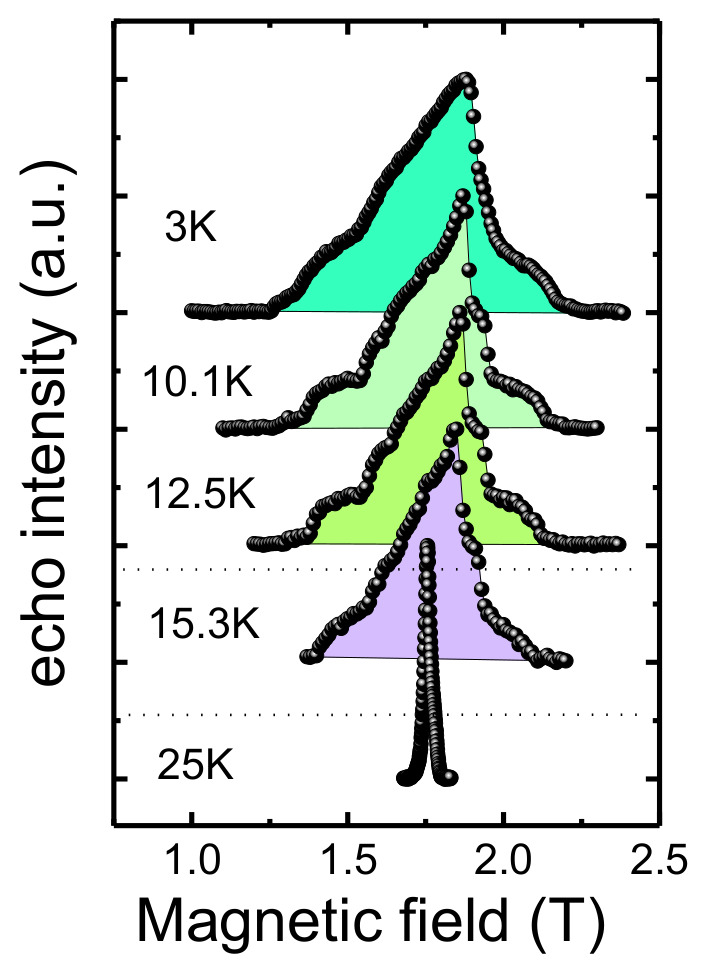
^7^Li NMR spectrum for LiMn_2_TeO_6_ at low temperatures in different external field. Dotted lines mark the phase boundaries (see paragraph 4).

**Figure 10 materials-15-08694-f010:**
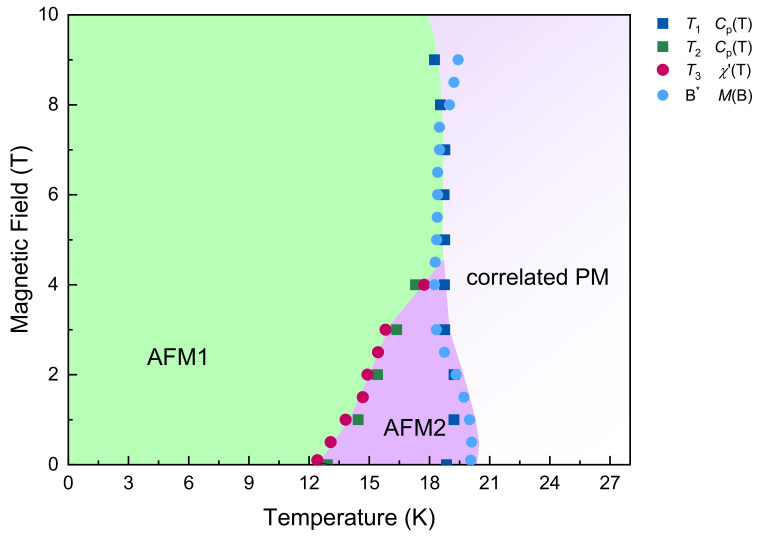
The magnetic phase diagram for tellurate LiMn_2_TeO_6_.

## Data Availability

Not applicable.

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
