# Peer review of "Static and Resonant Properties and Magnetic Phase Diagram of LiMn2TeO6"

_materials, 2022, doi:10.3390/ma15238694_

Round 1

Reviewer 1 Report

In this paper, a comprehensive study of thermodynamic and magnetic resonance properties of the lithium manganese tellurate LiMn2TeO6 was reported. It is an interesting content, but arranged structure needs to be improved. Therefore, it needs revision before it is published in this journal.

1. Authors can re-think about the title of the manuscript.

2. The abstract should be rewritten, i.e., it should highlight the originality, innovation and uniqueness of this paper.

3. More details about the “Experimental” section should be provided.

4. It is suggested to compare the results of the present research with some similar studies which is done before.

5. Extensive English revisions for language and grammar are strongly needed.

6. The conclusions should directly present the quantified research achievements, such as important models, data and indicators, etc. List them using serial numbers.

Author Response

Reviewer 1

In this paper, a comprehensive study of thermodynamic and magnetic resonance properties of the lithium manganese tellurate LiMn2TeO6 was reported. It is an interesting content, but arranged structure needs to be improved. Therefore, it needs revision before it is published in this journal.

  1. Authors can re-think about the title of the manuscript.

New title “Static and resonant properties, complex magnetic order and magnetic phase diagram of LiMn2TeO6” reflects the conducted research better.

  1. The abstract should be rewritten, i.e., it should highlight the originality, innovation and uniqueness of this paper.

The abstract is fully rewritten, it reflects the main findings of our research.

  1. More details about the “Experimental” section should be provided.

Preparation and identification of the sample was described in detail in Ref. 23. This description took 49 lines in Ref. 23, not to mention crystal structure determination that confirmed the sample composition. There is no need to repeat all of this in the new manuscript, whereas reporting only part of this will provide incomplete, and, therefore, distorted impression.

To emphasize the sample identity, we substitute, at the beginning of Section 2, the phrase “The preparation of LiMn2TeO6 powder samples by conventional solid-state reaction and their redox analysis has been described earlier [23]” with the following: “The preparation of LiMn2TeO6 powder samples by conventional solid-state reaction and their phase analysis, redox analysis and crystal structure determination have been described in detail earlier [23]. Here, we used samples from the same work, stored in a tightly closed container in a desiccator. Their identity was confirmed by X-ray diffraction.

  1. It is suggested to compare the results of the present research with some similar studies which is done before.

The results of the present study are compared with similar study on the iso-elemental compound Li2MnTeO6.

  1. Extensive English revisions for language and grammar are strongly needed.

These days, there are no native English speakers here, but we applied every effort to improve the presentation.

  1. The conclusions should directly present the quantified research achievements, such as important models, data and indicators, etc. List them using serial numbers.

The results of research are given in the Summary section of the revised manuscript.

Reviewer 2 Report

Comments:

1.      Some of the experimental results should be discussed in the abstract.

2.      Novelty of the work should be highlighted.

3.      Authors need to mention the details of the chemical suppliers in the materials section.

4.      It would be very useful to present the steps of reactions mentioned at the section 2 as a scheme.

5.      XPS study should be provided to show the existence of elements in the sample.

6.      Photoluminescence study should be provided to explain the existence of crystal defects in the sample.

7.      The discussion is too simple and not deep enough. More close connections and quantitative results should be established and discussed.

8.      The quality of the figure and figure captions should be improved.

9.      The discussion section should be expanded to highlight the scientific contribution of this study to this field

10.  The conclusion (summary) should be provided alone with more informative and show only significant findings of the study.

11.  Social implications shall be highlighted in the conclusion

12.  English is suggested to be polished by a native speaker to a publishable level.

13.  It is suggested the authors to check their manuscript carefully and thoroughly to avoid some typical mistakes and mistypes

Author Response

Reviewer 2

  1. Some of the experimental results should be discussed in the abstract.

In revised manuscript, we mention that the title compound orders magnetically in two steps at T1 = 20 K and T2 = 13 K. The intermediate phase at T2 < T < T1 is fully suppressed by magnetic field µ0H of about 4 T. The frequency dependence of ac magnetic susceptibility points to cluster spin glass-type behavior below T2. Beside magnetic phases transitions firmly established in static measurements a relaxation-type phenomena were observed well above magnetic ordering temperature in resonant measurements.

  1. Novelty of the work should be highlighted.

As written in the abstract, the combination of static and resonant methods of measurements, i.e., ac and dc magnetic susceptibility χ, magnetization M, specific heat Cp, electron spin resonance (ESR) and nuclear magnetic resonance (NMR), provides supplementary information on the various aspects of magnetic subsystem in LiMn2TeO6 including two-step ordering and spin-glass-type behavior.

  1. Authors need to mention the details of the chemical suppliers in the materials section.

See below.

  1. It would be very useful to present the steps of reactions mentioned at the section 2 as a scheme.

See below.

  1. XPS study should be provided to show the existence of elements in the sample.

Comments 3, 4, and 5 all refer to preparation and composition of the sample. Therefore, we provide a combined response to these three comments.

Preparation and identification of the sample was described in detail in Ref. 23 including reagent supplier, reagent qualification (“special purity” or “pure”), their drying, details of the solid-state synthesis, determination of Li:Mn:Te ratio by XRD phase analysis of variable compositions, and determination of oxygen content by redox titration. This description took 49 lines in Ref. 23, not to mention crystal structure determination that confirmed the sample composition. There is no need to repeat all of this in the new manuscript, whereas reporting only part of this will provide incomplete, and, therefore, distorted impression. As for XPS study, determination of the elemental composition is necessary when the sample is grown from a solution, or melt, or vapor. However, in our solid-state preparation, the Li:Mn:Te ratio in the starting mixture was set by accurate weighing, with uncertainty in the fourth decimal digit, i.e., much more accurate than with XPS (or EDX). Since no noticeable weight change was observed on heating the LiMn2O4+TeO2 mixture, its elemental composition was essentially the same as mixed, and only phase purity had to be verified by XRD. As noticed in Ref. 23, additional phase(s) appeared where weight gain was observed due to oxidation.

To emphasize the sample identity, we substitute, at the beginning of Section 2, the phrase “The preparation of LiMn2TeO6 powder samples by conventional solid-state reaction and their redox analysis has been described earlier [23]” with the following: “The preparation of LiMn2TeO6 powder samples by conventional solid-state reaction and their phase analysis, redox analysis and crystal structure determination have been described in detail earlier [23]. Here, we used samples from the same work, stored in a tightly closed container in a desiccator. Their identity was confirmed by X-ray diffraction.

  1. Photoluminescence study should be provided to explain the existence of crystal defects in the sample.

The main defect type supposed by the XRD structure determination is partial Mn3/Li2 site inversion. We do not expect that any photoluminescence study may confirm or exclude these defects.

  1. The discussion is too simple and not deep enough. More close connections and quantitative results should be established and discussed.

We rewrote the discussion adding information on the similar study performed on the iso-elemantal compound Li2MnTeO6.

  1. The quality of the figure and figure captions should be improved.

We applied efforts to improve the quality of the figures and figure captions. In particular, Figure 10 has been fully redrawn.

  1. The discussion section should be expanded to highlight the scientific contribution of this study to this field.

See reply to remark 7.

  1. The conclusion (summary) should be provided alone with more informative and show only significant findings of the study.

In revised manuscript, the summary is rewritten as a separate section.

  1. Social implications shall be highlighted in the conclusion

It is difficult to discuss in the original scientific article the social consequences of a purely fundamental research devoted to the study of a complex metal oxide compound in extreme conditions of low temperatures and strong magnetic fields. However, we mentioned in the Introduction that the mixed valence manganese oxides exhibit colossal magnetoresistance effect and attract attention as prospective materials in lithium-ion industry. Besides, it has been found that they are capable of strong light adsorption in the solar spectrum range and may serve as magnetic sensors, spintronic and magnetocaloric devices.

  1. English is suggested to be polished by a native speaker to a publishable level.

These days, there are no native English speakers here, but we applied every effort to improve the presentation.

  1. It is suggested the authors to check their manuscript carefully and thoroughly to avoid some typical mistakes and mistypes

We checked our manuscript trying to avoid mistakes and mistypes. The changes in the manuscript are marked in color.

Round 2

Reviewer 1 Report

The manuscript's quality has been substantially improved. I recommend its acceptance for publication in its present form.

Reviewer 2 Report

The authors addressed each comment in detail and made appropriate changes. I recommend its publication in the present form.